# Predictive markers for the early prognosis of dengue severity: A systematic review and meta-analysis

Tran Quang Thach[1], Heba Gamal Eisa[2], AlMotsim Ben Hmeda [3], Hazem Faraj[3], Tieu Minh Thuan [4], Manal Mahmoud Abdelrahman [5], Mario Gerges Awadallah[6], Nam Xuan Ha[7], Michael Noeske [8], Jeza Muhamad Abdul Aziz [9], Nguyen Hai Nam[10], Mohamed El Nile[11], Shyam Prakash Dumre [1], Nguyen Tien Huy [12]*, Kenji Hirayama[1,12]*

1 Department of Immunogenetics, Nagasaki University, Nagasaki, Japan, 2 Faculty of Medicine, Menoufia University, Shebin El-Koum, Egypt, 3 Faculty of Medicine, University of Tripoli, Tripoli, Libya, 4 Faculty of Health Sciences, McMaster University, Hamilton, Ontario, Canada, 5 Medical School, Ain Shams University, Cairo, Egypt, 6 Faculty of Medicine, Alexandria University, Alexandria, Egypt, 7 Hue University of Medicine and Pharmacy, Hue, Vietnam, 8 American University of the Caribbean School of Medicine, Cupecoy, Sint Maarten, 9 College of Health Sciences, University of Human Development, Sulaimaniyah, Iraq, 10 Graduate School of Medicine, Kyoto University, Kyoto, Japan, 11 Ministry of Health, Zagazig, Egypt, 12 School of Tropical Medicine and Global Health, Nagasaki University, Nagasaki, Japan

* hiraken@nagasaki-u.ac.jp (NTH); tienhuy@nagasaki-u.ac.jp (KH)

**Data Availability Statement:** All relevant data are within the manuscript and its Supporting Information files.

## Abstract

### Background

Predictive markers represent a solution for the proactive management of severe dengue. Despite the low mortality rate resulting from severe cases, dengue requires constant examination and round-the-clock nursing care due to the unpredictable progression of complications, posing a burden on clinical triage and material resources. Accordingly, identifying markers that allow for predicting disease prognosis from the initial diagnosis is needed. Given the improved pathogenesis understanding, myriad candidates have been proposed to be associated with severe dengue progression. Thus, we aim to review the relationship between the available biomarkers and severe dengue.

### Methodology

We performed a systematic review and meta-analysis to compare the differences in host data collected within 72 hours of fever onset amongst the different disease severity levels. We searched nine bibliographic databases without restrictive criteria of language and publication date. We assessed risk of bias and graded robustness of evidence using NHLBI quality assessments and GRADE, respectively. This study protocol is registered in PROSPERO (CRD42018104495).

### Principal findings

Of 4000 records found, 40 studies for qualitative synthesis, 19 for meta-analysis. We identified 108 host and viral markers collected within 72 hours of fever onset from 6160 laboratory-confirmed dengue cases, including hematopoietic parameters, biochemical

**Funding:** The author(s) received no specific funding for this work.

**Competing interests:** The authors have declared that no competing interests exist.

substances, clinical symptoms, immune mediators, viral particles, and host genes. Overall, inconsistent case classifications explained substantial heterogeneity, and meta-analyses lacked statistical power. Still, moderate-certainty evidence indicated significantly lower platelet counts (SMD -0.65, 95% CI -0.97 to -0.32) and higher AST levels (SMD 0.87, 95% CI 0.36 to 1.38) in severe cases when compared to non-severe dengue during this time window.

## Conclusion

The findings suggest that alterations of platelet count and AST level—in the first 72 hours of fever onset—are independent markers predicting the development of severe dengue.

### Author summary

The major concern in dengue fever is the abrupt occurrence of severe complications, for which only close monitoring of patients is the treatment scheme. Thus, the markers managing to predict the subsequent progression of complications—in the early stage of disease course—could alleviate the clinical management burden. Ideally, the predictors foretell the outcomes before the severe complications occur—usually on days 4–7 following fever onset. In this study, therefore, we reviewed the available markers collected during the first 3 days of fever onset. We found robust evidence of significantly lower platelet counts and higher AST levels in those who subsequently developed severe dengue than those who did not. In this regard, platelet count could serve as an independent warning sign rather than combining with hematocrit—as seen in the current classification—which remains unaltered during this time window. Also, abdominal pain and vomiting could predict the outcomes, but using these signs is arduous when their manifestations vary as per the patient without cutoffs. Hepatomegaly rate is substantially higher in severe dengue, but likely yields a high false-negative prediction rate. There is a need for larger studies to confirm the relatedness of hyaluronan in severe dengue.

## Introduction

Dengue fever is an acute mosquito-borne viral disease caused by infection of any of four dengue virus serotypes (DENV1–4) that predominantly circulates in tropics and subtropics, subjecting over 3 billion individuals to the risk of infection [1]. DENV accounts for an annual occurrence of ~ 400 million cases across 129 countries [2], though only ¼ is symptomatic [3].

Dengue clinical manifestation varies greatly from self-limiting febrile illness to fatal outcomes without clear-cut hallmarks to assist diagnosis. These life-threatening complications occur relatively late during the disease course—often day 4 of fever onset or around the critical phase [4,5]. At present, no therapeutics are available for dengue except supportive care as an off-label approach. Furthermore, dengue vaccine has acquired specific achievements, but on the horns of a dilemma, restricting the vaccinations only to those with dengue-infection history. Therefore, dengue management continues to rely upon constant examination and round-the-clock nursing care, imposing a burden on clinical triage and economy in resource-limited settings [1].

Representing a potential breakthrough in the proactive management of dengue, the effort to some extent has shifted to the search for means that can foretell the outcomes at the inception of disease. In 2009, the World Health Organization (WHO) revised dengue case classification in light of multi-centre study findings known as "DENCO" (DENgue COntrol), which was proved more sensitive to predicting severe cases [6,7]. The adapted classification, although improved, has demonstrated limited performance in the early prognosis of severe complications [8,9]. Given the betterment of pathogenesis understanding, accumulating evidence has associated numerous predictive candidates with severe dengue progression [10,11].

Nevertheless, the reported evidence is conflicting [12–14]. These conflicts arise from insufficient study power or inconsistency in the timeframe during which markers are recorded [15,16], often late in the disease course [15,16]. Additionally, the combination of dengue hemorrhagic fever (DHF) and dengue shock syndrome (DSS) as a severe form is frequently seen [14,17–19], which could mislead clinical triage given that management strategies differ between these two groups [20,21].

Previous systematic reviews offer insight as to predictive methods for severe dengue. Notwithstanding, the conclusions went hand in hand with the study limitations that formed the body of evidence, say the combination of DHF and DSS or late measurement [12,14,19,22,23]. Besides, studies have focused concern on immunogenetic markers that benefit pathophysiology rather than medical case management [18,24–27]. Here we advance the findings of prior work to illustrate the association of available markers and severe dengue in a different context.

## Methods

### Protocol registration

We previously developed and registered the systematic review protocol in PROSPERO (CRD42018104495; S1 Protocol). We followed the PRISMA reporting guidelines (S1 Table).

### Eligibility criteria

We included observational studies that reported on the association of host markers and dengue severities categorized by WHO classifications. The markers were measured within 72 hours of fever onset and before the occurrence of any severe complication, including shock, bleeding, and organ impairment.

### Search strategy

On 7th June 2018, we searched for articles using nine databases, including Cochrane Library, Clinicaltrials.gov, EMBASE, Google Scholar, POPLINE, PubMed, Scopus, SIGLE, and Web of Science (ISI). We updated the systematic search before the inception of data analysis (on 17th December 2019; without POPLINE database as no longer available) and before the submission for peer review (on 20th September 2020) by using the identical search terms (S1 Text). We manually searched preprint sites (bioRxiv and medRxiv), in-text citations from eligible articles, and previous reviews on the related topics. Concerning articles with insufficient information, we contacted the authors.

### Study selection

We screened titles and abstracts for the relevance of the content that was continued by reviewing full texts. Three reviewers independently worked on tasks with an agreement reached through a discussion amongst the reviewers, and in case of discrepancy, we consulted the

empirical authors (KH, NTH, SPD). We used the Kappa statistic to appraise the inter-rater reliability amongst the reviewers.

## Data extraction

We piloted the in-house data collection tool for its applicability before the official extraction by three reviewers working independently (S1 Data). We collected data for study characteristics, population baseline characteristics, measurement times, pre-admission treatments, dengue case classifications, dengue serotypes, serostatus (primary or secondary infection), and host marker data collected in the pre-specified time window.

For data reported by days before fever subsided, we included data reported within 3 days before defervescence, depending on the availability of data and the similarity of timeframe varying from study to study, assuming that day 4 was the day fever subsided [4,5,10,28]. Accordingly, we chose data from day 1–3 when authors reported data corresponding to the day since fever onset. For data graphically presented, we requested data from authors or used a web-based software program, available at https://automeris.io/WebPlotDigitizer/, to collect the summary information of the outcomes or individual participant data by which the latter was then checked for normal distribution and normalized before computing mean and standard deviation (SD). For missing SD, we outsourced data from other articles based upon the similarity of population, measurement time, and severe outcomes that patients developed [29]. If neither of these methods satisfied, we considered synthesizing the evidence by narrative review in tabular formats. Concerning the overlapping data, we chose outcomes with a larger sample size for meta-analysis.

## Quality assessment

At the study level, we used NHLBI quality assessment and Q-genie scoring to rate the risk of bias of the component studies corresponding to their designs, case-control or cohort [30] or genetic association studies [31] and presented them in the characteristics of included studies table. Briefly, each study underwent a set of signalling questions about the potential bias that a study may present. We mainly based the appraisal on the resemblance of the population, including age structures, enrollment times, and locations alongside the clear sample size justification. When the reviewers could not provide sound judgement due to insufficient information, the study was of unclear risk of bias.

At the outcome level, we used the GRADE approach to grade the certainty of our findings for their clinical applicability [32] and to generate an evidence profile, including the judgements on the risk of bias, inconsistency, imprecision, and indirectness of the findings [33–36]. We graded down the robustness of evidence when serious concern arose from any of the four domains.

## Outcomes and definition of markers

Outcomes were the differences in host marker data between severe and non-severe dengue cases. To uniform the severity levels through the series of WHO dengue classifications, to those of the 1997 classification or earlier, we grouped dengue fever, dengue hemorrhage fever grade 1 and grade 2 into non-severe cases; grade 3 and grade 4 (dengue shock syndrome) were defined as severe cases. Similarly, the 2009 WHO classification or later, we combined dengue with and without warning signs to form a group of non-severe cases.

The markers were biochemical substances (e.g., liver enzymes, VEGF), hematopoietic parameters (e.g., leukocyte, neutrophil), immune mediators (e.g., chemokines, cytokines), and

viral footprints (e.g., viral load, NS1 antigen detected in any of host tissue or biological fluid), or clinical symptoms or signs, which altered or occurred due to dengue infection.

### Statistical analysis

To ensure the appropriateness of conducting a meta-analysis, we initially examined the similarity across the studies based upon three dimensions, including clinical traits, methodology and observed effects [37,38]. Next, we performed a meta-analysis of the relationship between patient-derived data and severe dengue development using the Sidik-Jonkman method for a random-effect model that bears an adequate error rate in estimating the between-study variance [39]. We used Hedges's g—a standardized mean difference (SMD)—as the pooled estimates for continuous variables [40] and natural logarithm odd ratios (LORs) for binary outcomes [41], followed by 95% confidence intervals (CIs). For the articles that were of unmet similarity, we performed the narrative review in tabular formats.

As the rule of thumb, the statistical approaches of heterogeneity that we incorporated into the principal analysis ($I^2$, $\tau^2$, and Q test) reflected the arithmetical variability in estimates and the overlapping in confidence intervals [33]—rather than either actual clinical or methodological differences; therefore, we did not mainly base the exploration of the inconsistency on these approaches but the examination of clinical key features by GRADE approach. However, we could not fully perform the subgroup analysis and meta-regression to see the impact of the a priori hypotheses such as age, gender, pre-admission treatments, dengue case classifications, serotypes, serostatus, and study limitations (risk of bias) due to sporadic reporting data. We estimated mean and SD using the methods published elsewhere [42–44].

## Results

The systematic search identified 4000 records through the three different operations of search against time. After removing duplicates, 2666 records were included for the title and abstract screening. The manual search identified eight additional articles. We updated the systematic search and found two research articles. In total, 40 articles were utilized to generate the body of evidence. Of these, 19 articles underwent meta-analysis. For the remaining articles, we mainly focused on the central findings in tandem with the methodological flaws, as presented in Fig 1.

Kappa statistics showed that the strength of agreement between the reviewers at any cross-checked screening step was moderate, by which the index varied from 0.42 to 0.56, with 95% CI varied from 0.30 to 0.65.

The 40 studies involved approximate 6160 laboratory-confirmed dengue patients who met our pre-specified criteria from three endemic continents of dengue: Asia (especially South-East Asia countries), Latin America, and the Pacific Islands. Of these, 19 studies reported data on children; 11 included all age groups; eight included only adults; two did not describe the target population (Table 1).

At the study level, risk of bias varied from low to high. Of the 40 studies, 57.5% were of a high risk of bias, 30.0% and 12.5% had low and unclear risk, respectively. We performed an additional assessment concerning the genetic association study indicating a good study design (Table 1). The robustness of evidence was extremely low to moderate based on GRADE scoring (Table 2).

Overall, 14 studies assessed 15 hematopoietic parameters, the meta-analyses of four eligible parameters indicated significantly lower platelet counts in those who subsequently developed severe dengue than those who did not (n = 3671, SMD -0.65, 95% CI -0.97 to -0.32; Fig 2). By

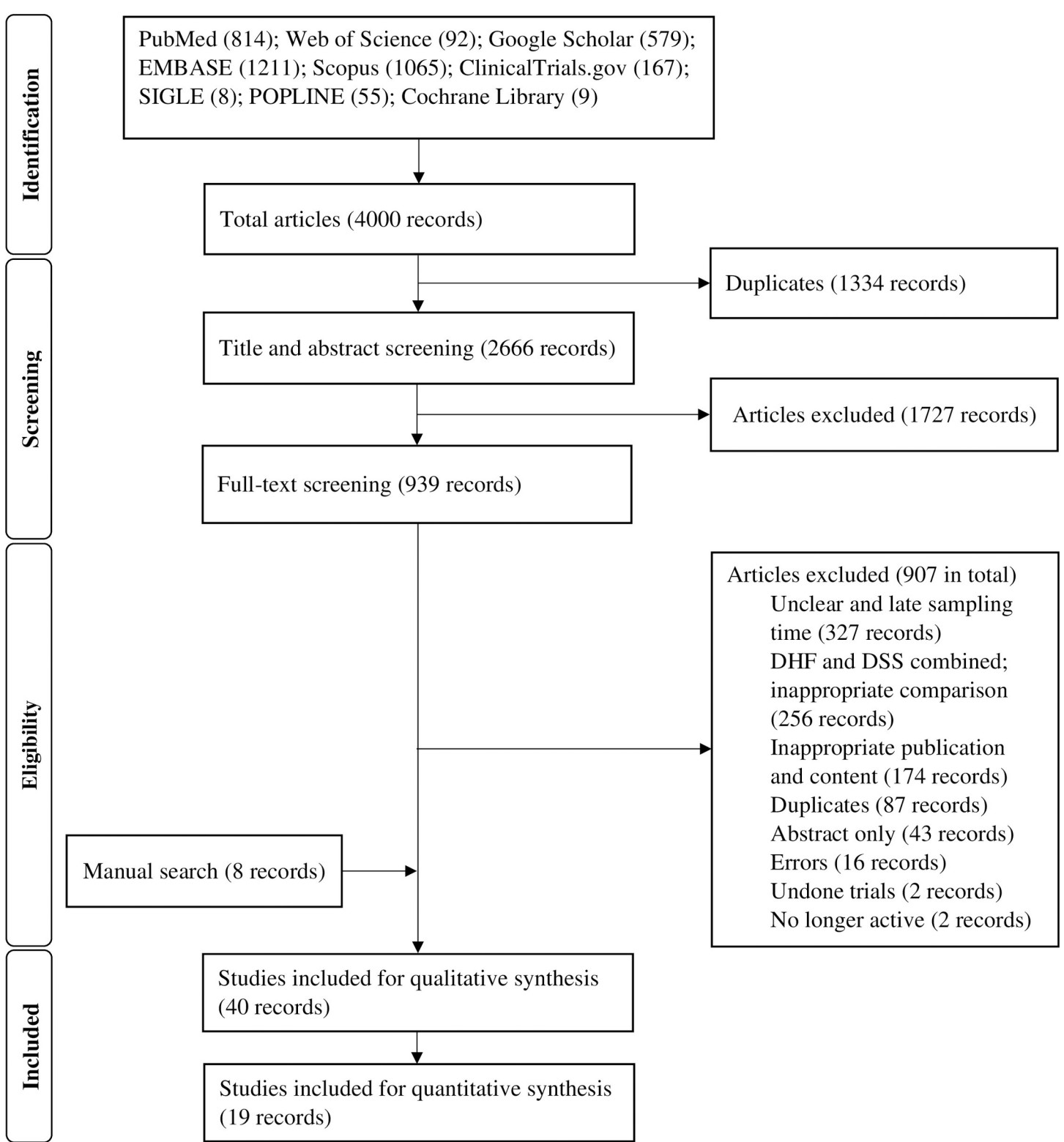

**Fig 1. PRISMA flow diagram of study selection.**

**Table 1. Characteristics of included studies.**

| | Method | Age* | Time† | Markers | Severe | Non-severe‡ | Outcomes | Comments on findings | Risk of bias |
|---|---|---|---|---|---|---|---|---|---|
| Aguilar et al., 2014, Mexico [45] | Prospective | 25 ± 15 yrs | Day 2–3 after fever onset (or day -2, -1 before fever subsided) | Viral load | NR | NR | 2009 WHO classification | Although viremia's kinetic changes showed significant differences between SD and NSD during the entire course of illness, there was no association between viremia and illness severity in the early stage | High |
| Avirutnan et al., 2006, Thailand [46] | Prospective | 9.6 ± 3.0 yrs | Day -3, -2 before fever subsided | Viral load, NS1, SC5b-9 | NR | NR | 1997 WHO classification | The alterations of viral load and NS1 levels were undifferentiated between shock and non-shock cases. On the contrary, SC5b-9 levels were specific to the illness severity, significantly higher in those with shock | Low |
| Biswas et al., 2015, Nicaragua [47] | Prospective | 64% of children aged from 5–12 yrs | Day 2–3 after fever onset | Total serum cholesterol, LDL, HDL | 38–77 | 34–137 | 1997 and 2009 WHO classification | On day 2 after fever onset, LDL-C level in SD was significantly lower than in NSD | Low |
| Butthep et al., 2006, Thailand§ [48] | Retrospective | 10.6 ± 3.7 yrs | Day -3, -2 before fever subsided | WBC, platelet counts, sTM, sVCAM-1, sICAM, sE-selectin, CECs, AAL, ANC, ALC | 1–3 | 7–21 | 1997 WHO classification | sTM was significantly higher in DSS patients than in DF and DHF patients during 3 days before fever subsided | High |
| Butthep et al., 2012, Thailand [49] | Retrospective | NR | Day -2 before fever subsided | IL-4, IL-6, IL-8, IL-10, IFN-γ, TNF-α, MCP-1, IL-1β, IL-2, VEGF, EGF, platelet counts | NR | NR | 1997 WHO classification | The alteration of most markers was not specific to DSS on this day. Notably, IL-6 level differentially increased in DSS as compared to non-DSS cases | Unclear |
| Chaiyaratana et al., 2008, Thailand [50] | Prospective | 11.0 (9.0–13.0) yrs | Day 3 after fever onset | Serum ferritin levels | 1 | 5 | 1997 WHO classification | Ferritin levels increased proportionally to dengue severity. At the cutoff ≥ 1200 ng/mL, ferritin was likely to be a predictor of DHF | High |
| Chandrashekhar et al., 2019, India [51] | Prospective | 37.7% of the children < 5 yrs | Day 2–3 after fever onset | Serum neopterin level | 19 | 58 | 2009 WHO classification | Neopterin level in SD was significantly higher than in NSD | Low |
| Chen et al., 2015, Taiwan§ [15] | Retrospective | 52.6 ± 16.0 yrs | Day 1–3 of after fever onset | CRP | 4–10 | 87–93 | 1997 and 2009 WHO classification | Regardless of the classifications, CRP levels successfully predicted the development of severe outcomes. For shock, at the cutoff of 30.1 mg/L, CRP yielded the sensitivity and specificity of 100% and 76.3%, respectively. For severe dengue, at the cutoff of 24.2 mg/L, the sensitivity and specificity was 70% and 71.3%, respectively | High |
| Chunhakan et al., 2015, Thailand§ [52] | Retrospective | 4–17 yrs | Day -2 before fever subsided | Platelet counts, IL-10, TNF-α, IL-1β | 2 | 16 | 1997 WHO classification | No association found | Unclear |
| Fernando et al., 2016, Sri Lanka§ [53] | Prospective | 32.3 ± 13.6 yrs | Day 3 after fever onset | AST, ALT, GGT, ALP, total bilirubin, albumin, IL-10, IL-17, viral load | 2 | 4 | 2009 WHO classification | ALP levels were slightly higher in SD than in NSD, which promptly returned to the normal range from day 4. IL-10 and IL-17 levels were likely to be associated with SD; however, the number of cases was inappropriate to see the effects | High |
| Hapugaswatta et al., 2020, Sri Lanka [54] | Retrospective | 28.5 in mean | Within the first 3 days of illness | The expression of the following microRNAs and putative target genes: let-7e, miR-30b, miR-30e, miR-33a, miR-150, EXH2, DNMT3A, RIP140, ABCA1 | 15 | 8 | 2009 WHO classification | miR-150 was highly abundant in SD as compared to that in NSD. At the cutoff of 7.54 ΔCq, miR-150 showed the good discriminative ability with a sensitivity of 80% and specificity of 88% | High |

(*Continued*)

**Table 1.** (Continued)

| | Method | Age* | Time† | Markers | Severe | Non-severe‡ | Outcomes | Comments on findings | Risk of bias |
|---|---|---|---|---|---|---|---|---|---|
| Hoang et al., 2010, Vietnam§ [55] | Prospective | 2–30 yrs | Less than 3 days after fever onset | Viral load, NS1 level, ANC, whole-blood transcriptional signature | 24 | 56 | Severe plasma leak according to the 2009 WHO classification | Neutrophil-associated CAMP and MPO plus the decoy receptor IL1R2 were differentially expressed between DSS and non-DSS patients. The findings suggested the association between neutrophil activation and the risk of shock in dengue. Plasma NS1 concentrations significantly increased in DSS | High |
| Hober et al., 1993, French Polynesia§ [56] | Retrospective | 3 mths–15 yrs | Day 1–3 after fever onset | Serum TNF-α, IL-6, IL-1β | 5–6 | 7–9 | 1975 WHO classification | TNF-α levels were indistinguishable amongst the severities; the highest value was observed in children with shock on day 3 of illness. IL-6 did not invariably increase; the highest values were seen in DHF1 on day 1, which sharply decreased on days 2 and 3, and were subsequently supplanted by DSS | High |
| Hober et al., 1996, French Polynesia [57] | Retrospective | 3 mths–15 yrs | Day 1–3 after fever onset | Serum sTNFR p75, TNF-α | 4 | 9 | 1980 WHO classification | sTNFR p75 increased in all the severity groups without marked differences. In addition, there was no relationship between sTNFR p75 and TNF-α levels | High |
| Koraka et al., 2004, Indonesia [58] | Retrospective | 7 mths–14 yrs | Day 2–3 after fever onset | sVCAM-1 | 10 | 22 | 1997 WHO classification | sVCAM-1 levels increased proportionally to the degrees of severity | Unclear |
| Kurane et al., 1991, Thailand§ [59] | Prospective | 4–14 yrs | Day -1 before fever subsided | sIL-2R, sCD4, sCD8, IL-2, IFN-γ | 3–5 | 4–10 | 1980 WHO classification | There were no striking differences in the proposed markers amongst dengue grades. By observation, DSS had lower IL-2 and IFN-γ levels than that in non-DSS dengue | High |
| Kurane et al., 1993, Thailand [60] | Retrospective | 5–14 yrs | Day -1 before fever subsided | IFN-α | 4 | 9 | 1986 WHO classification | No association found | High |
| Lam et al., 2017, Vietnam§ [61] | Prospective | 12 (10–13) yrs | Day 1–3 after fever onset | Vomiting, mucosal bleeding, abdominal pain, hepatomegaly, platelet counts | 80–81 | 1156–1186 | 1997 WHO classification | Patients who developed shock were likely to have lower platelet counts than those without shock, particularly one day before shock. Platelet counts during this timeframe had poorly predictive value with an AUC of 0.68 | Low |
| Lee et al., 2012, Singapore [62] | Retrospective | 35 (27–43) yrs | Day 1–3 after fever onset | AST, ALT | NR | NR | 1997 and 2009 WHO classification | According to the 2009 WHO dengue case classification, AST and ALT were specific to SD, whereas this trend was no longer held when categorized by the 1997 WHO classification. Overall, AST and ALT yielded the low discriminative ability of the complications in dengue | High |
| Liao et al., 2015, China [63] | Retrospective | 39.2 ± 15.4 yrs | Day 1–3 after fever onset | Viremia titer, sVCAM-1, IL-6, TNF-α, IL-10, IFN-γ, IL-17A, sTNFR1 | NR | NR | 2009 WHO classification | sVCAM-1, IL-6, and TNF-α levels in SD were significantly higher than in NSD | High |
| Lin et al., 2019, Taiwan§ [64] | Prospective | 22–90 yrs | Day 1–3 after fever onset | Hyaluronan, WBC, platelet counts, albumin, AST, ALT | 15 | 93 | 2009 WHO classification | At the cutoff ≥ 70 ng/mL, hyaluronan level successfully differentiated DWS+ and SD from DWS-; however, the discriminative ability was limited with sensitivity and specificity corresponding to 76% and 55% | Low |

(Continued)

**Table 1.** (Continued)

| | Method | Age* | Time† | Markers | Severe | Non-severe‡ | Outcomes | Comments on findings | Risk of bias |
|---|---|---|---|---|---|---|---|---|---|
| Low et al., 2018, Malaysia [65] | Prospective | 31.7 ± 14.4 yrs | Day 1–3 after fever onset | PTX-3, VEGF | 2–10 | 2–51 | 2009 WHO classification | At the cutoff ranging from 19.03 to 50.53 pg/mL, VEGF levels on days 2 and 3 after fever onset successfully predicted the progression of SD with the sensitivity and specificity of at least 70% and 76.47%, respectively | Low |
| Mekmullica et al., 2005, Thailand [66] | Retrospective | 8.8 ± 3.5 yrs | Day 1–3 after fever onset | Serum and urine sodium | 6 | 43 | 1997 WHO classification | Sodium was significantly higher in shock patients than in those without shock | Unclear |
| Nguyen et al., 2016, Vietnam§ [67] | Prospective | 1–15 yrs | Less than 3 days after fever onset | Vomiting, abdominal pain, mucosal bleeding, WBC, platelet counts, albumin, AST, viremia titer | 117 | 1943 | 2009 WHO classification | Vomiting, platelets, and AST were significantly different between SD and NSD, which yielded a good discriminative ability with an AUC of 0.95 when combined with positive NS1 rapid test | Low |
| Pandey et al., 2015, India§ [68] | Retrospective | 70% of the participants ≤ 35 yrs | Day -3, -2 before fever subsided | Serum level and mRNA expression of the following cytokines: IL-8, IFN-γ, IL-10, TGF-β | 21–40 | 30–31 | 2009 WHO classification | IL-8 level in SD was higher than in NSD; however, there were no differences in the transcriptional expression of IL-8 between the two groups. Inversely, IFN-γ mRNA was highest in SD these days, yet serum IFN-γ was indistinguishable between the groups. IL-10 shared the similar pattern | High |
| Park et al., 2018, Thailand§ [69] | Retrospective | 9.0 ± 3.0 yrs | Day -3 before fever subsided | AST, ALT, WBC, RLC, albumin, platelet counts | 9 | 147 | 1997 WHO classification | No association found | High |
| Patil et al., 2018, India [70] | Prospective | 24 ± 5.8 yrs | Day 1–3 after fever onset | AnnexinV, platelet counts, RBC, platelet MPs, RBC MPs, activated endothelial cell-derived MPs | 1 | 59 | 2009 WHO classification | No association found | High |
| Phuong et al., 2019, Vietnam§ [71] | Prospective | 6–44 yrs, 65.6% of the participants ≤15 yrs | Day 1–3 after fever onset | Abdominal pain, vomiting, mucosal bleeding, hepatomegaly, cfDNA level | 8 | 53 | 2009 WHO classification | Plasma cfDNA in SD was significantly higher than in NSD. At the cutoff ≥ 36.85 ng/mL, cfDNA showed fair discriminative ability with sensitivity and specificity corresponding to 87.5% and 54.7%, respectively | High |
| Prasad et al., 2020, India [72] | Prospective | 72 (48–96) mths | Day 3 after fever onset | AST, ALT, ALP, GGT, albumin, total proteins | NR | NR | 2009 WHO classification | Amongst the markers, liver transaminases increased early in the first 3 days of the illness course, which were higher in SD than in NSD | Unclear |
| Sehrawat et al., 2018, India [73] | Prospective | NR | Day 2–3 after fever onset | INF-γ, IL-6, TNF-α | NR | NR | 2009 WHO classification | TNF-α level was significantly higher in SD than in NSD. For INF-γ, the difference between the severities was observed only on day 2 of the illness course | High |
| Sigera et al., 2019, Sri Lanka§ [74] | Prospective | 27.5 (20–40) yrs | Day 1–3 after fever onset | Hgb, WBC, platelet counts, ANC, ALC, AST, ALT, sodium, potassium, creatinine, CRP, total bilirubin | 10 | 76 | 2011 WHO classification | No associations found | Low |
| Soundravally et al., 2008, India [75] | Retrospective | 26–53 yrs | Day 3 after fever onset | MDA, TAS, PCOs, platelet counts | NR | NR | 1997 WHO classification | PCOs levels were successfully discriminated DSS from DF and DHF. Platelet counts and MDA levels in DSS were significantly higher than in DF but not DHF | High |
| Srichaikul et al., 1989, Thailand§ [76] | Retrospective | 5–14 yrs | Day -2 before fever subsided | Platelet counts, PTT, PT, TT, fibrinogen, FDP, ECL, FM, BTG, PF4 | 0–3 | 0–4 | 1986 WHO classification | No association found | High |

(Continued)

**Table 1.** (Continued)

| | Method | Age* | Time† | Markers | Severe | Non-severe‡ | Outcomes | Comments on findings | Risk of bias |
|---|---|---|---|---|---|---|---|---|---|
| Suwarto et al., 2017, Indonesia§ [77] | Prospective | 22 (18–30) yrs | Day 3 after fever onset | Syndecan-1, heparan sulfate, chondroitin sulfate, hyaluronan, Claudin-5, VE-cadherin | 23 | 80 | 2011 WHO classification | High levels of Syndecan-1 and Claudin-5 were strongly associated with the subsequent development of severe plasma leakage | Low |
| Trung et al., 2010, Vietnam§ [78] | Prospective | 22 (15–35) yrs | Day 1–3 after fever onset | AST, ALT | 4 | 81 | 2009 WHO classification | No association was found. Intriguingly, the findings indicated that co-infection of chronic HBV did not change the risk of SD, albeit the slight increase in ALT level | Low |
| Vaughn et al., 2000, Thailand [79] | Retrospective | 18 mths–14 yrs | Less than 3 days after fever onset | Viremia titer | NR | NR | 1997 WHO classification | High viremia titers during the first 3 days of fever onset were associated with severe dengue | High |
| Vuong et al., 2016, Vietnam§ [80] | Prospective | 14 (11–19) yrs | Less than 3 days after fever onset | Vomiting | 12 | 67 | 2009 WHO classification | Vomiting was more prevalent in SD than in NSD. At the cutoff of two episodes per day, the discriminative ability yielded high sensitivity, 92%, but low specificity, 52% | High |
| Vuong et al., 2020, multi-country study§ [81] | Prospective | 15 (10–25) yrs | Less than 3 days after fever onset | CRP, viremia titer, AST, ALT, albumin, CK, WBC, RNC, RLC | 28–38 | 984–1075 | 2009 WHO classification | Although high CRP level was suggestive of severe dengue, the variation of CRP levels between those with and without severe outcomes was not substantial | Low |
| Wills et al., 2009, Vietnam [82] | Prospective | 12 (7–14) yrs | Day 1–3 after fever onset | PT, APTT, fibrinogen, FDP, platelet counts | 14–26 | 156–212 | 1997 WHO classification | Although PT increased proportionally to the degrees of plasma leak, the association was weak and not discerned from non-dengue controls, while platelet counts were strongly associated with the extravasation | Low |
| Zhao et al., 2016, China [83] | Prospective | 46.0 ± 20.9 yrs | Day 1–3 after fever onset | Urine IgA level | 3 | 16 | 2009 WHO classification | No association found | High |

NR = not reported. SD = severe dengue. NSD = non-severe dengue. DSS = dengue shock syndrome. NS1 = non-structural protein 1. LDL = low-density lipoprotein protein cholesterol. HDL = high-density lipoprotein cholesterol. WBC = white blood cell. sTM = soluble thrombomodulin. sICAM-1 = soluble intercellular adhesion molecule-1. sE-selectin = soluble E-selectin. CECs = circulating endothelial cells. AAL = absolute atypical lymphocyte. ANC = absolute neutrophil count.

ALC = absolute lymphocyte count. IL = interleukin. IFN = interferon. TNF = tumour necrosis factor. MCP-1 = monocyte chemoattractant protein-1. VEGF = vascular endothelial growth factor. EGF = epidermal growth factor. CRP = C-reactive protein. AST = aspartate aminotransferase. ALT = alanine aminotransferase.

GGT = gamma-glutamyl transferase. ALP = alkaline phosphatase. sTNFR = soluble tumour necrosis factor receptors. PTX-3 = pentraxin 3. TGF = transforming growth factor. sVCAM-1 = soluble vascular cell adhesion molecule-1. RBC = red blood cell. MPs = microparticles. cfDNA = cell-free DNA. Hgb = hemoglobin.

MDA = malondialdehyde. TAS = total antioxidant status. PCOs = protein carbonyls. APTT = activated partial thromboplastin time. PT = prothrombin time.

TT = thrombin time. FDP = fibrinogen degradation products. ECL = euglobulin clot lysis time. FM = fibrin monomer. BTG = beta-thromboglobulin. PF4 = platelet factor 4. CK = creatinine kinase. RNC = relative neutrophil count. RLC = relative lymphocyte count.

*Age was presented in mean ± SD, median (IQR), and range. For Trung et al., 2010 and Wills et al., 2009, age was presented in median and 90% range.

†Some studies had observations longer than 3 days of the disease course, but we limited data reporting to the first 3 days only.

‡For data reported by individual markers or day of illness, we presented the number of participants ranging from minimum–maximum sizes.

§Studies for meta-analysis.

¶For a genetic association study, we performed the additional assessment yielding a single score of 49, in other words, a good study design.

contrast, there were no differences in leukocyte, lymphocyte, and neutrophil counts between those with and without severe dengue (S1, S2, and S3 Figs).

Thirteen studies examined 18 biochemical markers. However, only four markers were eligible for meta-analysis, which showed significantly higher AST levels in severe dengue than in non-severe dengue (n = 3610, SMD 0.87, 95% CI 0.36 to 1.38; Fig 3). No relationship was

**Table 2. GRADE evidence profile.**

| | Studies and participants | Assessment of the body of evidence | Effect (95% CI) | Certainty**** | Importance†********* |
|---|---|---|---|---|---|
| Platelet counts | Seven observational studies. Severe cases (n = 236); non-severe cases (n = 3435) | Most information derived from the low-risk-of-bias studies, say 57%. The point estimate remained unchanged when removing studies with higher or unclear risk of bias, SMD –0.76, 95% CI -1.15 to -0.36, $I^2$ 82.41%. Several studies had different age structures or case classifications but did not alter the point estimates in subgroup analysis—children (n = 3477; SMD -0.59, 95% CI -1.00 to -0.17, $I^2$ 76.02%) versus adults (n = 194; SMD -0.78, 95% CI -1.30 to -0.26, $I^2$ 31.37%); the 1997 WHO classification (n = 1417; SMD -0.33, 95% CI -0.54 to -0.11, $I^2$ 0.74%) versus the 2009 WHO classification (n = 2254; SMD -0.98, 95% CI -1.34 to -0.61, $I^2$ 50.48%). Our a priori hypotheses could not well explain the heterogeneity than the excess of one large study effect (Nguyen et al., 2016); we, therefore, did not rate down for the inconsistency | SMD -0.65 (-0.97 to -0.32) | Moderate*** | Important***** |
| Aspartate transaminase (AST) | Seven observational studies. Severe cases (n = 195); non-severe cases (n = 3415) | Approximately 71% of the studies were of low risk of bias; no significant difference was noted when removing the high-risk-of-bias studies, SMD 0.93, CI 0.26 to 1.60, $I^2$ 91.62%. Even though the effect was stronger in children (n = 2216; SMD 1.47, 95% CI 0.10 to 2.85, $I^2$ 93.41%) than in adults (n = 1394; SMD 0.71, 95% CI 0.13 to 1.29, $I^2$ 4.01%), we noted no significant differences in point estimates. Six out of seven studies used the revised WHO classification, and the estimated effect remained unchanged in the sensitivity analysis, excluding a single study using the 1997 WHO classification, SMD 0.88, 95% CI 0.28 to 1.48, $I^2$ 95.10%. We did not rate down for the inconsistency, for the same reason as discussed for platelet counts | SMD 0.87 (0.36 to 1.38) | Moderate*** | Important****** |
| Abdominal pain | Three observational studies. Severe cases with events (55/206, 26.7%); non-severe cases with events (605/3178, 19.0%) | Abdominal pain characteristics vary from patient to patient, and we rated down one degree for the risk of bias as these characteristics were not considered effect modifiers. Although age structure was homogeneous across the studies, the different case classifications could result in variation between studies. However, we could not provide convincing statistical evidence due to a tiny number of studies; we conservatively rated down one degree for the inconsistency | lnOR 0.40 (0.01 to 0.80) | Very low* | Not important** |
| Vomiting | Three observational studies. Severe cases with events (135/209, 64.6%); non-severe cases with events (1254/3196, 39.2%) | Only one study accounted for the "dose-response" relationship between the clinical signs and severe dengue, and we rated down one degree for risk of bias. Also, we rated down one degree for inconsistency for the same reason as discussed for abdominal pain | lnOR 1.12 (0.37 to 1.87) | Very low* | Not important** |
| Liver enlargement | Two observational studies. Severe cases with events (4/89, 4.5%); non-severe cases with events (4/1221, 0.3%) | We rated down one degree for inconsistency as the included studies differed from age structures and dengue case classifications. Moreover, the optimal information size was unmet, as estimated to be approximately 300 events in a total sample to yield the precise point estimate, and we rated down one degree for the imprecision | lnOR 2.54 (1.11 to 3.96) | Very low* | Not important** |

(*Continued*)

**Table 2.** (Continued)

| | Studies and participants | Assessment of the body of evidence | Effect (95% CI) | Certainty**** | Importance†********* |
|---|---|---|---|---|---|
| Hyaluronan | Two observational studies. Severe cases (n = 38); non-severe cases (n = 173) | Although the two studies were clinically and methodologically uniformed, there were different magnitudes between study effects that our a priori hypotheses failed to explain, and we rated down one degree for the potential inconsistency. Although we found no data supporting the optimal information size in estimating serum hyaluronan effects in severe dengue, we conservatively rated down the imprecision by one degree for a sample size shorter than 400 | SMD 0.63 (0.21 to 1.05) | Very low* | Not important*** |

The certainty of evidence by GRADE approach. The participants were dengue laboratory-confirmed individuals presenting within 72 hours of fever onset. The outcomes were the differences in host markers between those who subsequently developed severe dengue and those who did not. We based the grading of risk of bias—at the outcome level—on the flawed measurements of the markers and the extent to which the individual study biases contributed to the inferences. We graded down the certainty by one degree for observational studies.

†We considered the findings important based on the current clinical landscape and the evidence certainty.

found between the alteration of ALT, albumin, and sodium levels with severe dengue (S4, S5, and S6 Figs).

Four studies monitored a total of four clinical signs in connection with the progression of severe dengue. The meta-analyses revealed the association of the presence of abdominal pain, vomiting, and liver enlargement with the increased risk of severe dengue (n = 3384, lnOR = 0.40, 95% CI 0.01 to 0.80; n = 3405, lnOR 1.12, 95% CI 0.37 to 1.87; n = 1314, lnOR = 2.54, 95% CI 1.11 to 3.96, respectively; Figs 4, 5, and 6). No relationship between mucosal bleeding and severe dengue risk was detected within this study (S7 Fig).

Seven studies proposed 13 host cell structure-associated markers. We could only estimate the effects of one marker, hyaluronan, which demonstrated significantly higher levels in those

| Study | Severe cases | | | Non–severe cases | | | Hedges's g with 95% CI | Weight (%) |
|---|---|---|---|---|---|---|---|---|
| | N | Mean | SD | N | Mean | SD | | |
| Chunhakan/2015/Thailand [52] | 2 | 80 | 49 | 16 | 92.13 | 36.23 | −0.31 (−1.79, 1.16) | 4.15 |
| Lam/2017/Vietnam [61] | 80 | 135.11 | 55.31 | 1,156 | 156.34 | 63.15 | −0.34 (−0.57, −0.11) | 24.87 |
| Lin/2019/Taiwan [64] | 15 | 90 | 49.18 | 93 | 137.03 | 46.46 | −1.00 (−1.56, −0.44) | 15.34 |
| Nguyen/2016/Vietnam [67] | 117 | 114.75 | 45.41 | 1,943 | 184.45 | 61.57 | −1.15 (−1.34, −0.96) | 25.81 |
| Park/2018/Thailand [69] | 9 | 158 | 68.3 | 147 | 183.43 | 89.06 | −0.29 (−0.96, 0.39) | 12.78 |
| Sigera/2019/Sri Lanka [74] | 10 | 121.96 | 73.53 | 76 | 154.88 | 62.91 | −0.51 (−1.17, 0.16) | 12.99 |
| Srichaikul/1989/Thailand [76] | 3 | 99.26 | 130.14 | 4 | 100.26 | 64.67 | −0.01 (−1.51, 1.49) | 4.05 |
| **Overall** | | | | | | | −0.65 (−0.97, −0.32) | |

Heterogeneity: $\tau^2$ = 0.10, $I^2$ = 68.43%, $H^2$ = 3.17

Test of $\theta_i = \theta_j$: Q(6) = 33.53, p = 0.00

Test of $\theta$ = 0: z = −3.91, p = 0.00

Favours non–severe cases    Favours severe cases

Random–effects Sidik–Jonkman model

**Fig 2. Forest plot showing the relationship between platelet counts and severe dengue.** One study was an outlier [67]; the estimated effects remained unaltered after the sensitivity analysis. The red dashed line represented the overall effect size.

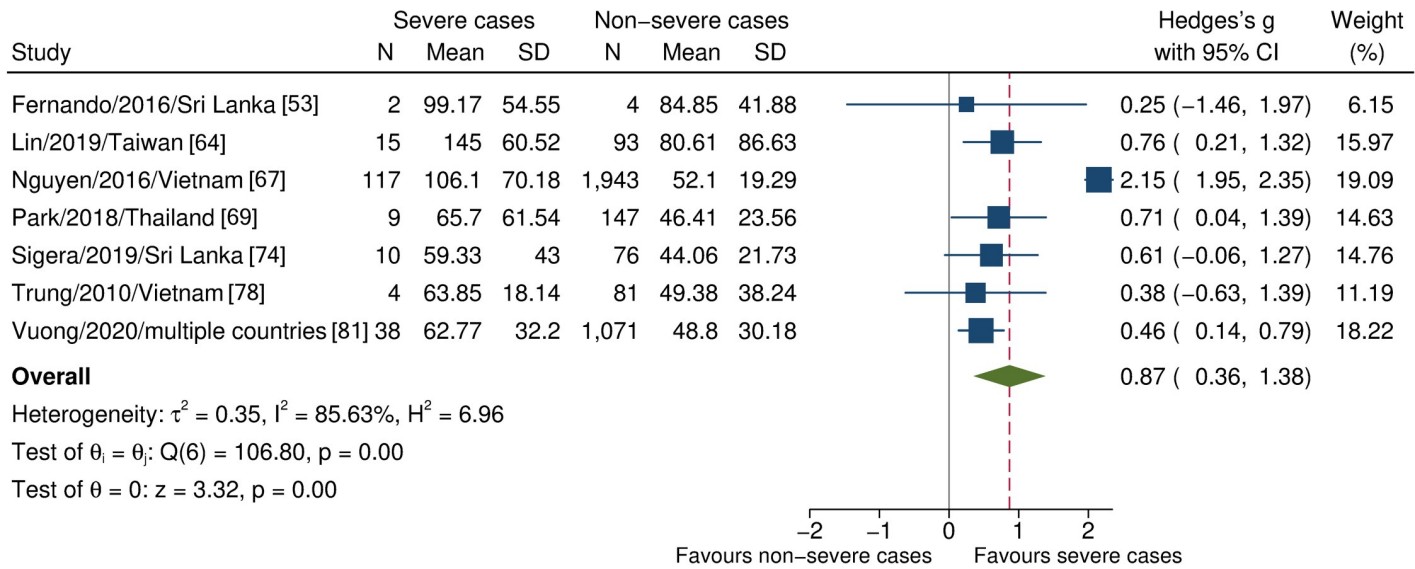

**Fig 3. Forest plot showing the relationship between AST levels and severe dengue.** One study was an outlier [67]; the estimated effects remained unaltered after the sensitivity analysis. The red dashed line represented the overall effect size.

who subsequently progressed to severe dengue (n = 211, SMD 0.63, 95% CI 0.21 to 1.05; Fig 7).

Seventeen studies reported the alterations of 25 immune mediators. Four eligible biomarkers, CRP, TNF-α, IL-10, and IFN-γ, found no significant variation in marker levels when comparing severe and non-severe dengue (S8–S11 Figs).

Eight studies correlated either the quantity of virus or NS1 antigen in the bloodstream with severity levels. The estimated effect pooled from three studies found no association between viral load in the early stage and the subsequent progression of severe dengue (S12 Fig).

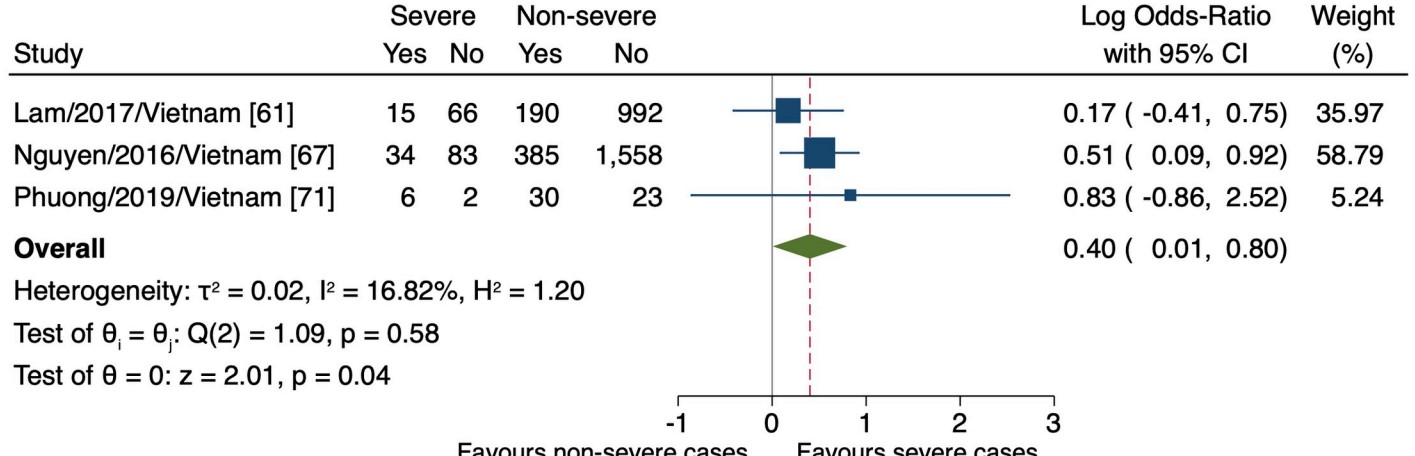

**Fig 4. Forest plot showing the relationship between the presence of abdominal pain and severe dengue.** The red dashed line represented the overall effect size.

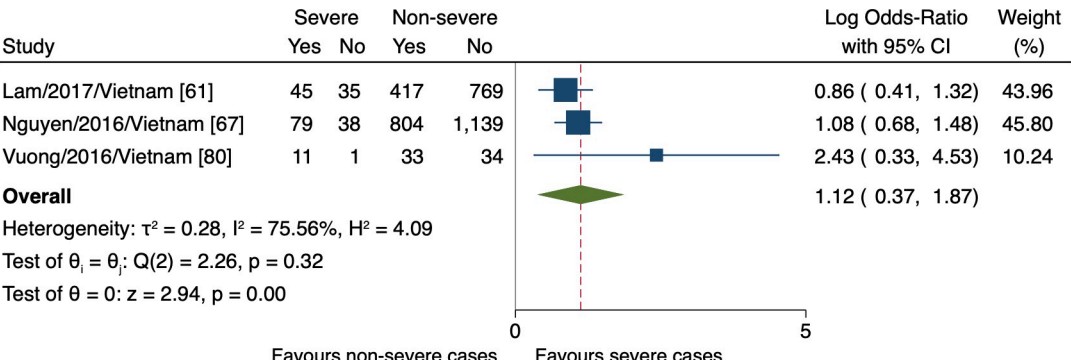

**Fig 5. Forest plot showing the relationship between vomiting and severe dengue.** The red dashed line represented the overall effect size.

Although statistical evidence implied the variation amongst the studies involving AST levels and platelet counts, the substantial heterogeneity ensued from the differences between small and large study effects rather than direction, which was apparent when we removed the most extensive study (Nguyen et al., 2016)—as was the outlier here—from the estimates (S13 and S14 Figs). On the other hand, other possible inconsistencies were not serious to compromise the estimated effects. In contrast, abdominal pain, vomiting, and enlarged liver showed low statistical heterogeneity despite the marked differences in the definition of severe outcomes.

## Discussion

This review found 40 studies comparing 108 host and viral markers amongst patients with varying dengue severity, published from 1989 to September 2020. Our findings suggested that the alterations of platelet counts and AST levels within 72 hours of fever onset were associated with severe dengue development. Similarly, the presence of abdominal pain, vomiting, liver enlargement and altered hyaluronan level were suggestive of the higher risk of severe dengue progression, but with exceptionally low robustness of the evidence.

Thrombocytopenia is commonly seen in dengue patients [84]. For this reason, platelet counts have long been used as a parameter to keep track of dengue progression. Our finding revealed an association between platelet counts and severe dengue, consistent with the previous systematic reviews [19,22], although we restricted the assessment to the first 72 hours

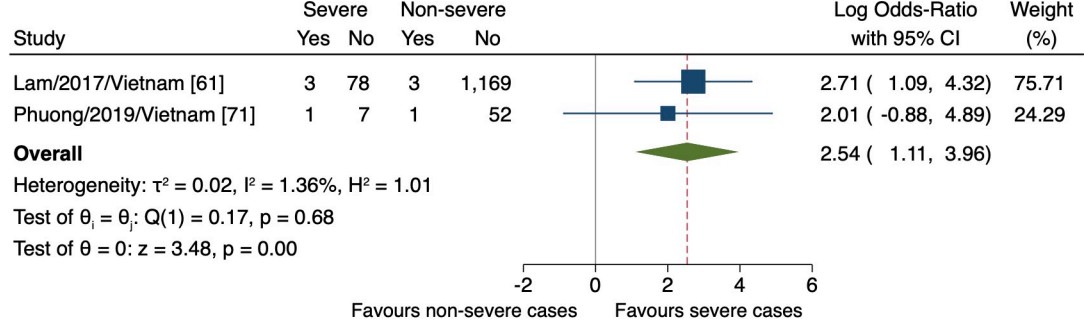

**Fig 6. Forest plot showing the relationship between hepatomegaly (>2 cm) and severe dengue.** The red dashed line represented the overall effect size.

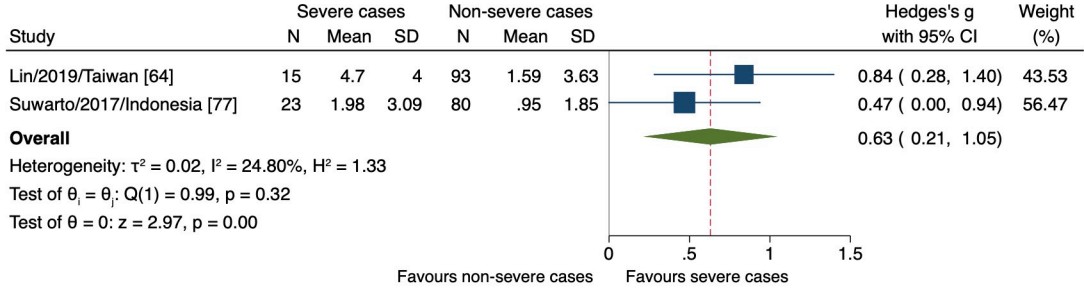

**Fig 7. Forest plot showing the relationship between hyaluronan levels and severe dengue.** The red dashed line represented the overall effect size.

following fever onset. The platelet decline occurs due to the massive activation of itself, apoptosis, and bone marrow hypoplasia, on which DENV initially has a direct or indirect impact [85–87]. In addition, the hyper-activated platelets, per se, could induce the extravasation by local secretion of pro-inflammatory mediators such as serotonin and VEGF [88,89].

As the findings have shown, AST level was significantly higher in severe than in non-severe patients during the early stage. Our analysis supported existing evidence of AST elevation in complicated dengue regardless of the time window [19,22,90,91]. Moreover, AST elevation alone was more indicative of systemic inflammation than hepatic injury, although DENV highly infects hepatocytes in the context of dengue tropism [92]. Generally, the elevation rate of AST is greater than that of ALT in dengue infection [93–97]. Wang et al., 2016 reported that 52% and 54% of mild and complicated dengue, respectively, demonstrated elevated ALT. When considering AST, these proportions increased to 75% and 80% [91]. The temporal change of liver transaminases begins early in the illness course, and elevations are significantly higher in severe dengue. Still, with moderate prediction power, especially in the case of ALT [62]. Instead, the combined index, such as $AST^2/ALT$, could improve discriminative performance [98]. On this point, our finding was inconsistent with most previous systematic reviews in which the ALT level was significantly higher in severe dengue [12,19,22]. This could be attributed to the time window that we used—the first 3 days versus 4 days or later in these studies—which could be premature for the hepatocellular damage to be noted.

We observed the association between abdominal pain, vomiting, liver enlargement, and altered serum hyaluronan level with severe dengue progression. However, our findings were unable to firmly confer their benefits in clinical practice due to weak evidence. Alternatively, our study puts forward several points that medical care may find helpful.

The major inconsistency in our findings regarding clinical signs was the existence of different case classifications. Given that the updated WHO classification, which includes broader clinical outcomes, is more sensitive to detecting severe cases than the 1997 guideline [6,99], the estimated effect of the markers defined was larger. This was apparent when comparing the effects of abdominal pain between two large cohorts, Nguyen et al., 2016 versus Lam et al., 2017, corresponding to the 2009 versus 1997 WHO classification (Fig 4). In the same way, vomiting was assigned more weights in Nguyen et al., 2016 study than that by Lam et al., 2017, but as yet, the difference was weaker (Fig 5).

Further, at the outcome level, bias and inconsistency may arise from the measurements of abdominal pain and vomiting. The effects could vary in terms of a "dose-response" relationship—referring to the resulting progressions of different clinical manifestations. Regarding vomiting, Vuong et al., 2016 suggested two episodes per day to predict severe dengue in general [80]; another study proposed three times per day associated with plasma leak [100]. Next,

many causes explain the acute abdominal pain in dengue, from non-specific to the more specific causes such as hepatitis, acalculous cholecystitis, pancreatitis, or several unusual causes [101–104], still having been merely referred to as "abdominal pain". It is clear that individuals who have a greater number of vomiting episodes are more likely to experience complicated dengue, and different clinical manifestations could speak to the different progressions. As such, vomiting and abdominal pain fulfill their prognostic tasks, but not optimally relax case-management pressure when using these signs—which ignores the beneficial cutoffs or hallmarks—goes with the umbrella admission. It also underscores the need to properly report clinical signs, featuring how the symptoms manifest—rather than whether they do present—in association with severe dengue.

There was evidence that hepatomegaly is more prevalent in complicated dengue [14,22,105–107]. Liver enlargement occurred in 1.0–34.6% of dengue infected adults [78,97,108–112]. The rate was even higher in children, 43.0–97.4% [107,110,113–119]. Nevertheless, the hepatomegaly rate was lower than expected in our study, despite the vast majority of participants being under 15 years old. One ultrasound study reported that 21.8% of children had an enlarged liver in the first 3 days of fever onset [120]. Based on this scenario, the optimal information size is approximately 300 events in total sample size, at the power of 80% and confidence level of 95%, to capture the real effects [34]. In comparison, the number of events in our analysis was shorter than the required size to provoke a precise point estimate. Despite this, the finding was consistent, underpinning the unclear liver involvement at the early stage of dengue infection—as no evidence of the substantial ALT and albumin differences between the clinical severities during this period.

The modest detection rate of hepatomegaly may require a re-evaluation of ultrasound benefits in the early stage, despite its proficiency in identifying sophisticated disturbances undetectable by physical examinations. Only few patients who subsequently developed complicated dengue exhibited fluid accumulation in the pleural cavity and peritoneal recesses (rectovesical pouch or pouch of Douglas), were reported during the first 72 hours of fever onset [121]. Gallbladder wall abnormalities became detectable on days 3–8 of the disease course [120–125]. Overall, the relationship between plasma leak signs—detected by ultrasound—and complicated dengue is undeniable. However, the sonographic hallmarks allow for reliable prediction mostly around the critical phase or later [121]. The plausible explanation, supported by Srikiatkhachorn et al., 2007, is that ultrasound requires a significant fluid accumulation to detect the differences. For this reason, although the differences between severe and non-severe dengue were detected during the first 3 days, a high false-negative prediction rate may occur. This explained the relatively low event rates in our study and the previous ones. To the best of our knowledge and as observed throughout this project, no evidence ascertains the performance of individual plasma leak signs by ultrasound during the first 72 hours of fever onset. The recent systematic review provided the broad landscape demonstrating a trade-off between sensitivity and specificity alongside the late presence of sonography signs or unclear measurement time [126].

Hyaluronan is the structural component maintaining the integrity of the extracellular matrix in connective tissues [127]. Hyaluronan increases during the inflammatory responses, reflecting the de novo synthesis and perturbed degradation that leads to its accumulation in the circulation [128]. However, few studies advance hyaluronan to explain dengue infection pathogenesis. Honsawek et al., 2007 first demonstrated the significantly increased hyaluronan level in children with DSS during the acute stage defined as days 3–7 [129]. Other studies noticed no differences in hyaluronan level between DHF and DF on day 3 of fever onset [77,130]. The different time windows and insufficient sample size appeared to render the

inharmonious conclusions. Thus, further studies with a larger size are needed to explore this association.

This study also has several limitations. First, the restrictive time window and "severe outcome" definitions yielded few studies as well as participants, which impacted in the capturing of the markers. Second, lacking data from Latin America and Africa—the frequent or continuous dengue risk areas—diminished our conclusions on the markers for these sites. Third, given that dengue patients reach the clinical outcomes do so by the multifactorial interactions, immune status and viral factors probably introduced noise into the findings. Nevertheless, such information was not always explicitly described for examining its potential impact on the inferences.

In conclusion, our review highlights the topics which merit further consideration. First, although the early alterations of platelets and AST levels indicate a higher risk of severe dengue development, these indicators require establishing quantitative diagnostic values and additional validation through prospective studies. Finally, decreased platelet counts in the first 72 hours could serve as an independent warning sign, instead of combining with elevated hematocrit detectable when plasma leak has implicitly occurred, often on day 3 or around the critical phase [61,74,131–133].

## Supporting information

**S1 Fig. Forest plot showing the relationship between leukocyte counts and severe dengue.** The red dashed line represented the overall effect size.
(TIF)

**S2 Fig. Forest plot showing the relationship between relative lymphocyte counts (RLCs) and severe dengue.** The red dashed line represented the overall effect size.
(TIF)

**S3 Fig. Forest plot showing the relationship between absolute neutrophil counts (ANCs) and severe dengue.** The red dashed line represented the overall effect size.
(TIF)

**S4 Fig. Forest plot showing the relationship between ALT levels and severe dengue.** The red dashed line represented the overall effect size.
(TIF)

**S5 Fig. Forest plot showing the relationship between albumin levels and severe dengue.** The red dashed line represented the overall effect size.
(TIF)

**S6 Fig. Forest plot showing the relationship between serum sodium and severe dengue.** The red dashed line represented the overall effect size.
(TIF)

**S7 Fig. Forest plot showing the relationship between the presence of mucosal bleeding and severe dengue.** The red dashed line represented the overall effect size.
(TIF)

**S8 Fig. Forest plot showing the relationship between CRP levels and severe dengue.** The red dashed line represented the overall effect size.
(TIF)

**S9 Fig. Forest plot showing the relationship between TNF-α levels and severe dengue.** The red dashed line represented the overall effect size.
(TIF)

**S10 Fig. Forest plot showing the relationship between IL-10 levels and severe dengue.** The red dashed line represented the overall effect size.
(DOCX)

**S11 Fig. Forest plot showing the relationship between IFN-γ levels and severe dengue.** The red dashed line represented the overall effect size.
(TIF)

**S12 Fig. Forest plot showing the relationship between viral load and severe dengue.** The red dashed line represented the overall effect size.
(TIF)

**S13 Fig. Sensitivity analysis showing the estimated effects of platelet counts.** The estimated effects remained unchanged by excluding seven studies; the heterogeneity considerably reduced by removing an outlier [67].
(DOCX)

**S14 Fig. Sensitivity analysis showing the estimated effects of AST levels.** The estimated effects remained unchanged by excluding seven studies; the heterogeneity considerably reduced by removing an outlier [67].
(DOCX)

**S1 Protocol. Study protocol.**
(PDF)

**S1 Table. Prisma checklist.**
(DOC)

**S1 Text. Search terms.**
(DOCX)

**S1 Data. Data extraction tool.**
(XLSX)

**S2 Data. List of excluded articles.**
(XLSX)

## Acknowledgments

I want to express my gratitude to all members of the Immunogenetic Department of Nagasaki University for the wholehearted advice at any study stage, in that data meetings were vital for me to keep this research on its right track. Further, I am thankful to the Online Research Club members (ORC) for the comprehensive remarks, especially Ms Simmies Ta, for her helpful conversations considering the language revision before submitting this research.

## Author Contributions

**Conceptualization:** Nguyen Tien Huy, Kenji Hirayama.

**Data curation:** Tran Quang Thach, Heba Gamal Eisa, Hazem Faraj, Tieu Minh Thuan, Manal Mahmoud Abdelrahman, Mario Gerges Awadallah, Nam Xuan Ha, Jeza Muhamad Abdul Aziz, Mohamed El Nile, Shyam Prakash Dumre, Nguyen Tien Huy.

**Formal analysis:** Tran Quang Thach, Heba Gamal Eisa, AlMotsim Ben Hmeda, Nguyen Tien Huy.

**Investigation:** Shyam Prakash Dumre, Nguyen Tien Huy, Kenji Hirayama.

**Methodology:** Tran Quang Thach, Tieu Minh Thuan, Shyam Prakash Dumre, Nguyen Tien Huy.

**Project administration:** Kenji Hirayama.

**Resources:** Nguyen Tien Huy.

**Software:** Tran Quang Thach, Nguyen Tien Huy.

**Supervision:** Nguyen Tien Huy, Kenji Hirayama.

**Validation:** Nguyen Tien Huy.

**Visualization:** Tran Quang Thach.

**Writing – original draft:** Tran Quang Thach, Michael Noeske.

**Writing – review & editing:** Tran Quang Thach, Nam Xuan Ha, Michael Noeske, Nguyen Hai Nam, Nguyen Tien Huy, Kenji Hirayama.

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
