## [Decision Letter · Decision Letter 0]

12 May 2021

Dear Dr. Huy,

Thank you very much for submitting your manuscript "Predictive markers for the early prognosis of dengue severity: a systematic review and meta-analysis" for consideration at PLOS Neglected Tropical Diseases. As with all papers reviewed by the journal, your manuscript was reviewed by members of the editorial board and by several independent reviewers. In light of the reviews (below this email), we would like to invite the resubmission of a significantly-revised version that takes into account the reviewers' comments. 

We cannot make any decision about publication until we have seen the revised manuscript and your response to the reviewers' comments. Your revised manuscript is also likely to be sent to reviewers for further evaluation.

Sincerely,

Melissa J. Caimano

Deputy Editor

Elvina Viennet

Deputy Editor

Reviewer's Responses to Questions

**Key Review Criteria Required for Acceptance?**

**Methods**

-Are the objectives of the study clearly articulated with a clear testable hypothesis stated?

-Is the study design appropriate to address the stated objectives?

-Is the population clearly described and appropriate for the hypothesis being tested?

-Is the sample size sufficient to ensure adequate power to address the hypothesis being tested?

-Were correct statistical analysis used to support conclusions?

-Are there concerns about ethical or regulatory requirements being met?

Reviewer #1: -Are the objectives of the study clearly articulated with a clear testable hypothesis stated? Yes.

-Is the study design appropriate to address the stated objectives? yes.

-Is the population clearly described and appropriate for the hypothesis being tested? yes.

-Is the sample size sufficient to ensure adequate power to address the hypothesis being tested? Not for every marker tested. However, considering the kind of analysis and that author stated this as a limitation of the study itself, it is acceptable to this reviewer.

-Were correct statistical analyses used to support conclusions? It seems to be. However, it should be better described. 

-Are there concerns about ethical or regulatory requirements being met? yes.

Overall, the methodology seemed solid, but it should be better described. It is a bit messy and not entirely clear. Statistical calculations and parameters are merely cited, which is ok. Still, it would be beneficial to count with a brief description of how they were calculated and what they represent. 

Also, three key points concern me most. In the first place, why did the authors just use 72 hours of fever as a cutoff for febrile phase analysis? This should be specified. It is not wrong, but for adults, it could be longer, usually between 2 and 7 days. Anyway, please provide the font where this information comes from or explain why you choose this particular time cut-off. Secondly, I still can't convince myself about grouping cases with mild dengue fever and cases with warning signs (or grade I/II of the 1997 classification) as a single non-severe group. Technically, these cases are not severe ones, ok. But they are supposed to be monitored, and sometimes they require hospitalization too. Indeed, warning signs are considered in some cases predictors of severe disease, even though this doesn't mean that their presence is mandatory for a severe outcome. So, perhaps this "warning signs" clinical category should be considered separately from the classical dengue fever and the severe cases. Finally, even though markers were analyzed independently, according to what I assume are the meta-analysis guidelines, I wondered if there is not possible to perform a multivariate meta-analysis. Dengue physiopathological basis is multifactorial. The patient's clinical outcome depends on the balance between the genetic and immunological background of the host and viral factors, as the infecting serotype, for example. Thus, even though the data presented here is interesting, it is no "new information" when checking each factor individually. Plus, the patient's immune status (primary vs. secondary infection) that is well known and described wasn't even approached here, nor the viral factors. It is hard to believe that there are no studies available that qualify for inclusion in this analysis. Also, I was wondering why so many studies from Latin America were excluded from your analysis. In this regard, I believe that authors should mention at least this imbalance because the host genetic background between Asian and Latin American populations might be different at some point and viral genotypes circulating in both continents. 

Minor comments

*Lines 154-155: Should not this be part of the search strategy?

*Line 160: be aware that Figure 1 is cited here for the first time, but its legend is in the results section. Please select where you consider it should appear for the first time. 

*Lines 164-167: Should not this be part of the study selection sub-section?

*Lines 170-171: Which criteria did the authors use for that assumption? Otherwise, please cite the corresponding reference. 

*Lines 181-184: Please cite the corresponding references. As well, a brief description of each procedure would be helpful to understand what is being done. 

*Lines 192-193: It is a strong claim. Be careful because many of them are not exclusive of dengue infection.

Reviewer #2: I do not have significant concerns regarding methodological aspects of this report (except that for a non-expert, these can look pretty complex).

**Results**

-Does the analysis presented match the analysis plan?

-Are the results clearly and completely presented?

-Are the figures (Tables, Images) of sufficient quality for clarity?

Reviewer #1: -Does the analysis presented match the analysis plan? yes

-Are the results clearly and completely presented? yes, but they could be improved. See my comments below. 

-Are the figures (Tables, Images) of sufficient quality for clarity? yes, but I would suggest some improvements, such as in the suppl tables, that are important but not clear. 

*Supplementary Information 4: Please update, and provide a shorter and clearer version. Remove unnecessary comments like "check again for sure". There are many sheets within this file, which make it hard to follow. The information presented in this file is of great relevance. 

*It wasn't clear to this reviewer how the Kappa statistics represented the reviewers' agreement, which the authors mentioned was moderate (lines 222-224). What does this mean in terms of study inclusion? Did the authors select studies without full agreement between the three of them? 

*It is still not completely clear to this reviewer how the calculated risk of bias was taken into account for further analysis (lines 270-272). 23 out of 40 initially selected studies presented a high risk, and some of them were included in the meta-analysis, right? Also, the "unclear" ones. Please provide a clearer explanation for this issue. Also, it is striking that the distinction between severe and non-severe cases wasn't available for some studies. How did the authors process that information? 

*It could be helpful maybe if studies included in the meta-analysis were somehow highlighted in Table 1. 

*Lines 286-292. Maybe this information could be easily read in a table. 

*As mentioned in the methodology section, it concerns how results are considered without age stratification (categorization or similar). Authors claimed that "Age was homogeneous across the studies—to a certain extent" (line 296). However, this is not what is observed in Table 1. On the other side, please consider avoiding ambiguous expressions like "to a certain extent" and provide precise information instead. 

Also, regarding the severe/non-severe classification: the authors followed the same rationale as previously published meta-analysis. Still, it is questionable if results could change by considering three rather than 2 categories. 

*Lines 309-310: "... instead of combining with hematocrit occurring relatively late around the critical phase" Please provide the corresponding reference.

*Lines 312-313: "... indicated that those with early low blood platelet counts were likely to develop severe dengue ...". In my opinion, authors should be careful about claims in terms of probabilities. This meta-analysis considers the qualitative presence/absence of a certain marker in non-severe and severe dengue cases. Though, the marker is being assessed with the clinical outcome already established. This is not a prospective study with patients being accompanied along time. Besides, even though it was discussed by the authors, platelet count and AST level are continuous variables here interpreted as qualitative, considered "low" and "elevated", respectively. What do the "low" and "elevated" represent, i.e., did the authors considered any particular cut-off? Again, these values would probably be different between adults and children, both considered indistinguishably in this analysis. 

*Line 314: Is SMD interchangeable with Hedge's g value presented in the figures? 

*Figures: please include in the figures' legend the meaning of the red dashed line. 

*Lines 328-334: In line with what's mentioned above, abdominal pain, vomiting, and hepatomegaly are already considered warning signs by the WHO, which means that they are often related to severe cases. Cases presenting these signs/symptoms shouldn't, in my opinion, be considered mere "non-severe" cases. However, there is a known lack of consensus within the scientific community in this regard. Though, consider this just as an opinion and/or suggestion. 

*Lines 353-357: It is confusing whether the outlier study from Nguyen and collaborators was or was not removed since it appeared in figures, but it is mentioned in these lines that it was removed.

Reviewer #2: Results are clearly presented, with adequate set of figures and tables (and additional materials in appendices).

**Conclusions**

-Are the conclusions supported by the data presented?

-Are the limitations of analysis clearly described?

-Do the authors discuss how these data can be helpful to advance our understanding of the topic under study?

-Is public health relevance addressed?

Reviewer #1: -Are the conclusions supported by the data presented? yes

-Are the limitations of analysis clearly described? yes, but could be improved. 

-Do the authors discuss how these data can be helpful to advance our understanding of the topic under study? yes, but could be improved. 

-Is public health relevance addressed? yes

Specific comments: 

*Lines 367-368: "successfully foretold the impending severe outcomes, with exceptionally low robustness of the evidence" Be careful with this kind of claim because the presence of abdominal pain, vomiting, and hepatomegaly are indeed warning signs that may indicate a potential progression to severe disease, but it doesn't necessarily mean that patients presenting these signs/symptoms will do so. Plus, it is also mentioned that evidence robustness was low. 

*Line 369: "Clinically, bleeding is frequently present in dengue patients, irrespective of disease severity". Please provide the corresponding reference. 

*Line 374: replace "owing" for "due to".

*Line 374: "massive activation" of coagulation?

*Line 375: replace "of" for "on", and "has" for "have". 

*Line 375-376: Please provide the corresponding reference for this information. 

*Lines 376-377: "Additionally, it is worth noting that the platelet decline was bound up with not only the resultant hemorrhage but also the plasma leak". Again, be careful with this kind of claim because platelet decline is typical during the febrile phase, even in patients not progressing to severe conditions. Also, it would be relative to the level of platelets you might consider as normal, low, and so on.

*Lines 383-384: unclear sentence. 

*Line 402: "presented" not "present". 

*Lines 404-407: nor the 2009 classification consider them as severe dengue. 

*Lines 409: remove "with"

*Line 415: "remained" not "remains". 

*Line 422: replace "in speaking of" for "regarding".

*Line 424: replace "plain" for "clear". 

*Line 426: "fulfill" not "fulfils". 

*Lines 427-429: unclear sentence.

*Line 433: replace "from" for "in"

*Line 434: replace "in" for "of"

*Line 440: replace "fell far short" for "was shorter than"

*Lines 442-444: unclear sentence. 

*Lines 453-454: unclear sentence. 

*Line 455: which severity groups?

*Line 461: what do you mean by "late display"? The 72hs? 

*Line 462: "component" not "components". 

*Line 473: replace "other than" for "besides"

*Lines 474-475: What do you mean with "The evidence, needless to say, was weak and attached to the limitations of our study"?

*Line 477: replace " which was indispensable for capturing the underlying effects" for "which impacted in the capturing of the markers". 

*Lines 479-481: unclear sentence. 

*Line 483: replace "nonetheless" for "therefore". 

*Line 487: "indicated a higher" not "indicate the higher".

Reviewer #2: The conclusions are consistent with the results. Limitations are described and parallels with recent literature are adequately made.

**Editorial and Data Presentation Modifications?**

Reviewer #1: Overall, I would suggest carrying on a language revision. Even though the manuscript is not grammatically wrong-written, I found it hard to follow and challenging to understand due to its wordiness. Unfortunately, this aspect would negatively impact its reading and comprehension. Thus, it could somehow diminish the reader's interest in such an important topic. I would recommend being more concise, keep consistency in the verb conjugation, etc.

Some comments about the introduction section: 

*Line 113: DENCO abbreviation goes for...?

*Lines 120-126, 129-130, 130-131: Please provide the corresponding references for this information.

Reviewer #2: Minor comments:

Abstract: I would suggest to also mention in the abstract the markers like abdominal pain, vomiting, hepatomegaly and hyaluronan (in addition to platelets and AST, as it was done in the author summary).

Introduction:

- Lines 99-100: there is a confusion between DENV infection and dengue disease. The sentence could be rephrased as follows for accuracy: “DENV accounts for an annual occurrence of ~400 million infections across 129 countries, though only a ¼ is symptomatic”.

- Line 106: the main issue regarding the only vaccine currently available (Dengvaxia) is not suboptimal efficacy but that this is limited to people who have been previously exposed to dengue at least once (which means necessary screening) (as this vaccine was associated with an increased risk of severe dengue in vaccinated subjects without pre-exposure).

Methods:

- The Protocol (in appendix) says “empiric counsellor” and in the Methods, “empirical reviewers”. Can the role of such reviewers or counsellors be clarified?

Results:

- General comment: footnotes of some tables are very long but contain important information. Might some of those information be moved to main text?

- Line 222: is the moderate strength of agreement between the reviewers at cross-checked screening step an issue? Can it be clarified (or even discussed)?

- Study by Nguyen et al. 2016 is found to be an "outlier" for several marker, can possible cause(s) be discussed by the authors?

**Summary and General Comments**

Reviewer #1: This manuscript presented a review and meta-analysis aiming to identified host markers correlated with dengue disease severity. It addressed an extremely important topic with a high impact on public health. Previously published meta-analyses demonstrated the potential effect of several viral and host markers in predicting severe dengue disease. The analyses here presented focused particularly on the early prognosis, considering factors within the 72 initial hours since the onset of symptoms. However, there are some issues regarding the methodology and analysis itself that need to be improved.

Reviewer #2: This article by Thach et al. presents the outcome of a metanalysis that focused on predictive markers of severe dengue (especially those collected within 3 days of fever onset). The authors have considered initially 4000 articles to only retain at the end 40 and 19 studies for the qualitative and quantitative assessments respectively (taking into account 108 potential markers).

This is a well-written report. The methodology is transparent but sometime pretty complex for a non-expert of meta analyses.

Results are in a way disappointing as level of evidence is pretty low and direct impact on clinical practice is still far away (no specific cutoff for instance for platelet drop). This is not necessarily surprising considering context of dengue and the works done so far toward predictive markers of dengue severity. But the results are well-described, and makes sense. They are also fully discussed in the context of relevant literature. 

This work is of interest for dengue community as it highlights a limited set of predictive markers of severe dengue of high potential, and encourages further research in this direction. It also spots discrepancies in data reporting, the use of different guidelines which underline the need for more consistency and standardization.

PLOS authors have the option to publish the peer review history of their article (what does this mean?). If published, this will include your full peer review and any attached files.

Reviewer #1: No

Reviewer #2: No
---

## [Decision Letter · Decision Letter 1]

16 Aug 2021

Dear Dr. Huy,

Thank you very much for submitting your manuscript " Predictive markers for the early prognosis of dengue severity: a systematic review and meta-analysis" for consideration at PLOS Neglected Tropical Diseases. As with all papers reviewed by the journal, your manuscript was reviewed by members of the editorial board and by several independent reviewers. The reviewers appreciated the attention to an important topic. Based on the reviews, we are likely to accept this manuscript for publication, providing that you modify the manuscript according to the review recommendations. 

We appreciate your patience with this lengthy process. I know the manuscript has been under consideration for an extended period of time. But, please understand that the goal of the Review process is to improve the quality of the work, both in execution and presentation. Please review carefully the suggestions of Reviewer 1 when considering how to revise and resubmit.

Sincerely,

Melissa J. Caimano

Deputy Editor

Elvina Viennet

Deputy Editor

We appreciate your patience with this lengthy process. I know the manuscript has been under consideration for an extended period of time. But, please understand that the goal of the Review process is to improve the quality of the work, both in execution and presentation. Please review carefully the suggestions of Reviewer 1 when considering how to revise and resubmit.

Reviewer's Responses to Questions

**Key Review Criteria Required for Acceptance?**

**Methods**

-Are the objectives of the study clearly articulated with a clear testable hypothesis stated?

-Is the study design appropriate to address the stated objectives?

-Is the population clearly described and appropriate for the hypothesis being tested?

-Is the sample size sufficient to ensure adequate power to address the hypothesis being tested?

-Were correct statistical analysis used to support conclusions?

-Are there concerns about ethical or regulatory requirements being met?

Reviewer #1: -Are the objectives of the study clearly articulated with a clear testable hypothesis stated? yes

-Is the study design appropriate to address the stated objectives? yes

-Is the population clearly described and appropriate for the hypothesis being tested? yes

-Is the sample size sufficient to ensure adequate power to address the hypothesis being tested? Not completely for some analyses, but it has been pointed out by the authors as a limitation. 

-Were correct statistical analysis used to support conclusions? yes

-Are there concerns about ethical or regulatory requirements being met? yes

Reviewer #2: (No Response)

**Results**

-Does the analysis presented match the analysis plan?

-Are the results clearly and completely presented?

-Are the figures (Tables, Images) of sufficient quality for clarity?

Reviewer #1: -Does the analysis presented match the analysis plan? yes

-Are the results clearly and completely presented? yes

-Are the figures (Tables, Images) of sufficient quality for clarity? yes

Reviewer #2: (No Response)

**Conclusions**

-Are the conclusions supported by the data presented?

-Are the limitations of analysis clearly described?

-Do the authors discuss how these data can be helpful to advance our understanding of the topic under study?

-Is public health relevance addressed?

Reviewer #1: -Are the conclusions supported by the data presented? yes

-Are the limitations of analysis clearly described? yes

-Do the authors discuss how these data can be helpful to advance our understanding of the topic under study? yes

-Is public health relevance addressed? yes

Reviewer #2: (No Response)

**Editorial and Data Presentation Modifications?**

Reviewer #1: 53: vigilant doesn´t seem to be the proper word. You meant something like “constant”?

67: host and viral markers

68: dengue cases

70: viral particles and genes for viral factors?

71: Which analysis lacked statistical power, the quali or meta one?

81: markers allowing for predicting? Maybe markers managing to predict? 

83: should have told sounds strange. Maybe use foretell?

88: platelet count

97: Dengue virus

105: remedy for care

107: dengue-infection

113-115: …..DENCO Study Group findings (DENgue COtrol), which were proved more sensitive to predicting severe cases [6, 7]. Looks like something is missing after “(DENgue COtrol),”. Please check and rephrase. 

116: detection or prognosis?

117-119: references missing. 

120: these findings. Which findings? Please try to be more specific. Also, be careful with the use of terms like “issue” because it sound vague depending on the context. For exm, in line 129. Avoid using vague terms. 

120-134: This has been asked before in the previous revision. Please provide the corresponding references. 

122-124: DHF and DSS are both severe forms of the disease, and their combination is not to compensate for low sample sizes as proposed, is because they both represent severe outcomes. Please correct this information. 

126-127: Not clear what the authors mean by “Given the discrepancy in methods and

favored outcomes, the inconsistency in findings is inevitable”. Which methods, which favored outcomes, which findings?

139-141: Hard to read. 

175: would the be checked for was then checked. 

176: SD abbreviation is missing. Provide the corresponding abbreviations the first time they appear in the text. 

178: would consider for considered.

202: can be for were. 

204-205: (e.g., viral load, NS1 antigen detected in any host tissue or biological fluid), or clinical symptoms….

218: reflect for reflected.

233: By remarked you meant focused?

241: Latin America without s.

268-269: …. in which 57.5% of the studies were of a high risk of bias, 30.0% were of low risk of bias, and 12.5% had an unclear risk of bias. These percentages are over the 40 selected studies or the 19 included in the meta-analysis? Please clarify. 

281-284: This looks more a like a discussion. Consider moving it to the next section. 

304: signs or symptoms. 

309-310: Tough claim. Consider toning it down by using “was detected within this study” instead of “exist”. 

319: to severe dengue

336: to compromise

341: host and viral markers

361-364: hard to read. 

363: Be careful, not all non-structural proteins. NS1 is the predominantly one, and is highly immunogenic. On the other side, it’s not that it targets hepatocytes. DENV directly infects this cell type. 

365: the previous study….. Which one?

365-366: Reference is missing. 

371-373: This looks inconsistent with what stated above, cited in ref 36, whose authors didn’t detect a difference in Alt o Ast levels between mild and severe dengue. Percentages are almost the same between both categories. 

373: Reference is missing.

374: has still been for which could be. 

378-379: Hard to read. 

380: review for study. 

380: researchers or medical care? 

381: findings with final s

383-387: For this and other reasons is that classification was updated in 2009. 

388: had for were.

389-390: misuse of dashes. 

392: abdomen for abdominal

393-394: Not clear at all which study uses which classification. 

394-395: 1.43 vs 1.69. Is this a relevant difference? Doesn´t look like. 

399: dose-response relationship? Not clear what the authors mean by that. 

404: forms of…. ??

407: different clinical manifestations could speak to the different aetiologies with the corresponding progressions. Not clear. What do you mean by different aetiologies? Should it not be the same, i.e., dengue infection? 

408-414: Hard to read. Consider splitting into shorter sentences with more conciseness. 

416: remove than in others. 

418: meagre? 

421: figure for scenario. 

421-423: not clear. 

424-425: the finding was consistent with our observations. Your observations are not your findings? Seems redundant. 

435: By this period you mean the 72 first hours?

452: since Honsawek et al., 2007 to observe the raised hyaluronan level in children with DHF [76]. Please rephrase. 

454: inferences or differences? Inferences mean conclusions. 

451-466: It can be seen that the information is there, but it is hard to read. Please rephrase, considering splitting long sentences into shorter and more concise ones. 

467-469: unclear. Please rephrase. 

470: impact without final s. 

474: Second for Finally.

Reviewer #2: (No Response)

**Summary and General Comments**

Reviewer #1: The study presented by Thach et al is a systematic review and meta-analysis that looked for host and viral markers potentially predicting a progression of dengue into a severe outcome. Improvements and corrections in the manuscript could be identified. 

I am still concerned about two topics: 

1) The fact that studies' selection wasn't under full agreement of the three researchers. Instead they use a moderate Kappa estimator. This means that some sotudies wouldn't have been chosen by one of the three reviewers. 

2) The difficulties found to somehow merge both WHO dengue classifications (1997 and 2009) are understandable, however, as pointed out in my previous revision, it is tricky to consider cases with warning signs as mere mild dengue cases. They are not severe ones, indeed. But how sure could we be that cases published as warning signs didn't progress to severity? On the other side, the authors mentioned in the text that ws (for exm abdominal pain, vomiting, and liver enlargement) could serve as predictors for severe outcome (which is not new), but the sample size was too low. I found that contradictory in a way of saying, considering the answers given to the previous revision. Anyway, the authors made their point and were ok. I just would be very careful with conclusions obtained from this study since groups, mostly the non-severe dengue one, included very wide clinical outcomes. 

Finally, I would strongly recommend asking for a language revision. The text is full of typos and not concise or clear at many points. Otherwise, this would, unfortunately, impact the quality of the work here presented and be less attractive to readers.

Reviewer #2: All my questions and comments have been adequately addressed by the authors.

PLOS authors have the option to publish the peer review history of their article (what does this mean?). If published, this will include your full peer review and any attached files.

Reviewer #1: No

Reviewer #2: No

Figure Files:

Data Requirements:

Reproducibility:

References

---

## [Editor Report · Decision Letter 2]

10 Sep 2021

Dear Dr. Huy,

We are pleased to inform you that your manuscript 'Predictive markers for the early prognosis of dengue severity: a systematic review and meta-analysis' has been provisionally accepted for publication in PLOS Neglected Tropical Diseases.

Best regards,

Melissa J. Caimano

Deputy Editor

Elvina Viennet

Deputy Editor

---

## [Editor Report · Acceptance letter]

28 Sep 2021

Dear Dr. Huy,

We are delighted to inform you that your manuscript, " Predictive markers for the early prognosis of dengue severity: a systematic review and meta-analysis," has been formally accepted for publication in PLOS Neglected Tropical Diseases.

Best regards,

Shaden Kamhawi

co-Editor-in-Chief

Paul Brindley

co-Editor-in-Chief
